# New insight on some selected nanoparticles as an effective adsorbent toward diminishing the health risk of deltamethrin contaminated water

Samar M. Ibrahium[1]*, Ahmed A. Farghali[2], Rehab Mahmoud[3], Ahmed A. Wahba[4], Saeed El-Ashram[5,6]*, Hesham A. Mahran[7,8], Shawky M. Aboelhadid[9]

1 Animal Health Research Institute, Fayoum Branch, Fayoum, Egypt, 2 Materials Science and Nanotechnology Department, Faculty of Postgraduate Studies for Advanced Sciences (PSAS), Beni-Suef, University, Beni-Suef, Egypt, 3 Department of Chemistry, Faculty of Science, Beni-Suef University, Beni-Suef, Egypt, 4 Parasitology Department, Animal Health Research Institute, Dokki, Egypt, 5 College of Life Science and Engineering, Foshan University, Foshan, Guangdong, China, 6 Faculty of Science, Kafrelsheikh University, Kafr El-Sheikh, Egypt, 7 Health Informatics Department, College of Public Health and Tropical Medicine, Jazan University, Jazan, Saudi Arabia, 8 Hygiene, Zoonoses and Epidemiology Department, Faculty of Veterinary Medicine, Beni-Suef University, Beni-Suef, Egypt, 9 Department of Parasitology, Faculty of Veterinary Medicine, Beni-Suef University, Beni-Suef, Egypt

* Drsamarmahmoud333@yahoo.com (SMI); elashram@yahoo.com (SEA)

**Data Availability Statement:** All relevant data are within the paper.

**Funding:** The author(s) received no specific funding for this work.

## Abstract

Deltamethrin is a widely used insecticide that kills a wide variety of insects and ticks. Deltamethrin resistance develops as a result of intensive, repeated use, as well as increased environmental contamination and a negative impact on public health. Its negative impact on aquatic ecology and human health necessitated the development of a new technique for environmental remediation and wastewater treatment, such as the use of nanotechnology. The co-precipitation method was used to create Zn-Fe/LDH, Zn-AL-GA/LDH, and Fe-oxide nanoparticles (NPs), which were then characterized using XRD, FT-IR, FE-SEM, and HR-TEM. The kinetic study of adsorption test revealed that these NPs were effective at removing deltamethrin from wastewater. The larval packet test, which involved applying freshly adsorbed deltamethrin nanocomposites (48 hours after adsorption), and the comet assay test were used to confirm that deltamethrin had lost its acaricidal efficacy. The kinetics of the deltamethrin adsorption process was investigated using several kinetic models at pH 7, initial concentration of deltamethrin 40 ppm and temperature 25˚C. Within the first 60 min, the results indicated efficient adsorption performance in deltamethrin removal, the maximum adsorption capacity was 27.56 mg/L, 17.60 mg/L, and 3.06 mg/L with the Zn-Al LDH/GA, Zn-Fe LDH, and Fe Oxide, respectively. On tick larvae, the results of the freshly adsorbed DNC bioassay revealed larval mortality. This suggests that deltamethrin's acaricidal activity is still active. However, applying DNCs to tick larvae 48 hours after adsorption had no lethal effect, indicating that deltamethrin had lost its acaricidal activity. The latter result corroborated the results of the adsorption test's kinetic study. Furthermore, the comet assay revealed that commercial deltamethrin caused 28.51% DNA damage in tick cells, which was significantly higher than any DNC. In conclusion, the NPs used play an important role in

**Competing interests:** The authors have declared that no competing interests exist.

**Abbreviations:** D, Deltamethrin; LDH, layered double hydroxide; XRD, X-ray diffraction; FTIR, Fourier transforms infrared spectroscopy; HRTEM, High resolution transmission electron microscope; FESEM, field emission high resolution Scanning Electron Microscope; DNPs, Deltamethrin/ nanoparticles; NPs, Nanoparticles; DNCs, Deltamethrin nanocomposites.

deltamethrin decontamination in water, resulting in reduced public health risk. As a result, these NPs could be used as a method of environmental remediation.

## Introduction

Deltamethrin (D) is a pyrethroid pesticide that is commonly used to control household and crop pests. Its widespread use has resulted in serious environmental and public health issues. Deltamethrin binds to sodium channels, causing hyperexcitability in neurons, paralysis, and death in insects [1, 2]. Deltamethrin can cause reproductive toxicity, nerve damage, and chronic disease in humans (long-term exposure), as well as being harmful to the environment [3]. Deltamethrin acute toxicity was investigated in Amazonian fish species, with the results revealing high toxicity ($LC_{50}$-96 h values ranging from 6.69 to 23.63 μg/L) [4]. Exposure to hazardous drug discharge via water stream during agricultural irrigation has a negative impact on water quality. The extensive use of herbicides in agriculture has been linked to negative effects on the environment and aquatic organisms (such as oxidative stress, genotoxicity, neurotoxicity, and immunotoxicity) [5–7]. Residual pesticides are a major cause of many diseases, including cancer, birth defects, and severe health effects. Ozone gas ($O_3$) was used to remove leftover deltamethrin, which was utilized as an insecticide in the treatment of wheat crops (*Triticumaestivum* L.) [8]. Additionally, soil is a very effective degrading agent for deltamethrin elimination (95 percent in 54 min at pH 10) in cotton field water [9]. Deltamethrin was removed (88.3 and 82.8 percent, respectively) by degrading its residuals using two *Serratiamarcescens* (DeI-1 and DeI-2, respectively) bacterial strains [10]. Atrazine has a detrimental effect on the silver catfish, crayfish, and common carp's liver metabolism and immunity [11–13]. B-glucan (BG) was utilized to mitigate the adverse impact of pendimethalin (PMN) on the liver, kidney, and immunological response of *Oreochromisniloticus* [14], and B-glucan supplementation before atrazine exposure provided significant protection against atrazine-induced water pollution damage [15]. *Rhipicephalus annulatus* is one of the most common ixodid ticks, and it's found all over the world in hot climates [16]. Ticks are a carrier for deadly illnesses, including *Babesia* and *Anaplasma* [17, 18], as well as a substantial economic loss owing to their impact on cow productivity [19]. In addition, synthetic chemical acaricides, such as deltamethrin, have been extensively employed to manage animal ectoparasites and domestic insect pests (mosquitoes, cockroaches, flies, and fleas) [20]. Additionally, the emergence of acaricide resistance has boosted the usage of these drugs, resulting in their use at higher dosages [21]. Pyrethroids may contaminate water and cause stunted development in aquatic creatures, as well as hepatic dysfunction, anti-oxidative imbalance, and immunosuppression in fish [7].

Nanotechnology is one of the most recent and excellent scientific fields, making major contributions to the advancement of human health. Nanoparticles (NPs) have been used in a variety of areas due to their distinct characteristics, which include form, high catalytic reaction, thermal conductivity, size distribution, and large surface area, all of which are desirable in a variety of applications [22]. Furthermore, they have been widely used in environmental restoration, contaminant removal, and clinical treatment [23]. The green synthesis of nanomaterials using plants, fungi, bacteria, and algae that have biomedical reagents is considered an environmentally friendly, biocompatible, cheap, and safe method [24]. Stabilizing Fe oxide is vital to prevent the agglutination and degradation of their chemical and physical properties in colloidal solutions. This has been achieved by coating Fe oxide using a coating agent, which often requires the addition of secondary reagents after reduction. Many academics have lately shown an interest in the elimination of organic contaminants using layered double hydroxide

(LDH). This is due to its distinct characteristics, which include a large surface area, low toxicity, and cheap cost, a high capacity for anion substitution, recoverability, and excellent chemical and thermal property stabilities [25]. Several methods for LDH modification have been described, including the rebuilding process, the ion exchange process, and co-precipitation in the presence of organics [26]. Many limitations of LDHs remain, including the inability to be used in extremely acidic or basic media. The goal is to prepare LDH materials utilizing innovative processes and sophisticated modifications, as well as ecologically friendly ways and simple operation. However, iron oxide (magnetic nanoparticles), which is considered one of the most safe and relatively simple nanoparticles to synthesize, has significant antimicrobial activity, as well as drug and gene delivery systems [27], and several therapeutic applications in anemia and cancer treatments [28]. We chose Zn–Fe LDH and Zn-Al LDH/GA as a model over other LDHs and Fe oxide because of their high stability constant and low solubility product [22]. Furthermore, the growing volume of solid adsorbent waste necessitates the development of novel recycling techniques. This is a crucial need all around the globe [29–32]. As a result of our review, we investigated the ability of Zn-Fe LDH, Zn-Al/GA LDH, and Fe oxide to remove deltamethrin and convert it to an inactive substance, thereby minimizing environmental pollution. This study highlights the potential application of various nanoparticles as an efficient deltamethrin adsorbent, broadening its scope for use in environmental remediation processes. Through a kinetic analysis of the adsorption test, we aimed to evaluate an adsorbate system with Zn–Fe LDH, Zn–Al LDH/GA, and Fe oxide used as suitable adsorbent materials. FT-IR, XRD, FE-SEM, and HR-TEM were used to characterize the produced materials.

## Materials and methods

### Ethical approve

The study was conducted under the roles of the ethical standards approved by Faculty of Veterinary Medicine, Beni-Suef University, Egypt and its specific approval number was (021–172). All experiments were performed in accordance with relevant guidelines and regulations.

### Used materials

For the synthesis of Zn-Fe LDH, Fe- oxide NPs, and Zn-Al-GA LDH nanoparticle-functionalized deltamethrin hybrids, commercial deltamethrin 5% (Butox®, EC; 5% active ingredient, Arab Company for Chemical Ind. Cairo, Egypt) (Table 1) was utilized. SDFCL, India, provided iron nitrate, Fe (NO3) 9H2O, and zinc nitrate, Zn (NO3)2•6H2O. Alpha Chemika, India, and Oxford Laboratory Reagent, India provided chloride salts, aluminum chloride (AlCl3), and zinc chloride (ZnCl2). Sigma-Aldrich provided hydrochloric acid (Carlo Erba reagents), sodium hydroxide (Biochem for Laboratory Chemicals in Egypt), and gallic acid. The Egyptian market provided the clove buds. Deltamethrin removal from wastewater was detected using Zn-Fe/LDH, Zn-Al-GA/LDH, and Fe oxide NPs. Furthermore, tick larvae were tested with deltamethrin, deltamethrin/Zn-Fe LDH, deltamethrin/Fe oxide NPs, and deltamethrin/Zn-Al-

**Table 1. Chemicals and physical characteristics of deltamethrin [33].**

| Pesticide name | Deltamethrin |
|---|---|
| Appearance | Off-white solid powder (technical grade) |
| Molecular formula | $C_{22}H_{19}Br_2NO_3$ |
| Molecular weight (g/mol) | 505.2 |
| Density (g/cm$^3$) | 0.550 |
| λ max (nm) | 290–385 |

GA LDH composites at various concentrations (recommended dosage (X) = 1uL/mL distilled water).

## Preparation of nanoparticles

**Zn-Fe LDH and Zn-Al-GA LDH nanoparticles.** The co-precipitation technique was used to make Zn-Fe LDH (4:1) and Zn-Al-GA LDH (4:1:1) nanoparticles. Zinc and iron nitrates were combined in 50 mL distilled water at room temperature in a 4:1 molar ratio. Sodium hydroxide (2 mol/L) was added drop by drop to pH 8.0 with continuous stirring for 24 h until the precipitation of Zn-Fe LDH was complete. For Zn-Al LDH/GA preparation, the same molar ratio (4:1:1) of Zinc and Aluminum chlorides and GA was used. The resulting precipitate was filtered and rinsed with distilled water multiple times at pH 7.0. For 24 h, the filtrate was dried in a vacuum oven at 50˚C [34, 35].

**Synthesis of iron oxide NPs from clove bud extract.** Five grams of dried grinding clove buds (Syzgyiumaromaticum) were washed twice with distilled water to remove dust before being combined with 250 mL of distilled water and boiled at 100˚C for 10 min. After cooling to room temperature, the extract was centrifuged and filtered using No. 1 Whitman filter paper. The filtrate was utilized to create Fe-oxide NPs. 15 mL of clove extract were combined with 5 mL of 0.3 molar iron nitrate (adjusted to pH6 with 0.1 molar NaOH) and incubated at room temperature for 10 h. The Fe NPs were centrifuged, washed three times with distilled water, followed by ethanol, dried at 40˚C, and stored for characterization. Green produced Fe oxide was calcined in an oven at 550˚C for 4 h under air [36].

**Characterization of nanoparticles.** To determine the structural composition of the synthesized nanocomposites, X-ray diffraction was performed on a PANalytical (Empyrean) X-ray diffraction with Cu-K radiation (wave length 0.154 nm) at an accelerating voltage of 40 kV, current of 30 mA, scan angle range of 5 to 80˚, and scan step 0:04˚. Fourier Transform Infrared Spectroscopy (FTIR) was performed on a PerkinElmer FTIR Spectrum BX PerkinElmer Life and Analytical Sciences, CT, USA, using KBr pellets in a 1:100 ratio and spectra recorded in the 400: 4000 wave numbers (cm-1) range to classify the binding groups present before and after adsorption of deltamethrin on the adsorbent surface of different nanoparticle vehicles. High resolution transmission electron microscope (HRTEM, JOEL JEM-2100) images with 200 KV as accelerated voltage and images of field emission high resolution Scanning Electron Microscope (Gemini, Zeiss-Ultra 55) images were used to determine the morphological characteristics and microstructure of nanoparticles (FESEM).

## Adsorption study

Adsorption tests were carried out to determine the effect of the produced nanomaterials on the applied deltamethrin. Falcon tubes (50 mL) contained 0.05 g of the produced adsorbent and 40 uL/mL of the pollutant deltamethrin. The pH of the solution was changed from 3 to 10 using HCl or NaOH (0.10 N), and measurements were taken using a Metrohm 751 Titrino pH meter. All tests were conducted in the dark, and the Falcon tubes were shaken for 48 hours at 250 rpm on an orbital shaker (SO330-Pro). After each adsorption operation, the catalyst was separated from the solution using syringe filters (Millipore Millex-G, 0.22 μm pore size). A UV–Vis spectrophotometer (UV-2600, Shimadz, Japan) was used to determine the residual concentration of deltamethrin at a wavelength of 250–385 nm [3, 37] at the start of preparation as well as after 1, 3, 24, and 48 hours. To ensure repeatability, all tests were carried out in triplicate. The amount of deltamethrin removed is estimated by (Removal percent) $Q\% = [(C_0 - C_t)/ C_0] \times 100$ as $C_0$ is the initial deltamethrin concentration and $C_t$ is the deltamethrin concentration at a time (t) [38]. The residual samples from the adsorption tests were collected and centrifuged to obtain residues containing the

Zn–Fe LDH/Delta, Zn–Al LDH/GA/Delta, and Fe oxide/Delta nanocomposites, which were then washed several times with twice distilled water and dried in an oven at 50˚C for 24 hours until completely dry. Equilibrium conditions were investigated by isotherm models and discussed in terms of nonlinear equations. We demonstrated the significance of our results using the statistical parameters $R^2$. Finally, we assessed the acaricidal effectiveness of these deltamethrin nanocomposites against *Rhipicephalus annulatus* larvae using dried reuses.

## Consistency of results and quality assurance

A UV–Vis spectrophotometer was used to measure the residual deltamethrin concentration in the samples. The study plastic and glassware were cleaned and rinsed in a 5% HCl aqueous solution before being immersed in bi-distilled water. The accuracy of deltamethrin records was evaluated by repeatedly introducing a deltamethrin solution standard into the UV–Vis spectro- photometer to obtain a calibration curve (R2 = 0.999). Three reference deltamethrin solutions were conducted after every 15 samples to ensure the spectrophotometer data was reliable.

To ensure repeatability, all experiments were repeated three times, and the average concen- tration was calculated using the mean and standard deviation (SD) (SPSS version 16). A statis- tically significant p-value was defined as less than 0.05.

## Tick collection and preparation of larvae for larval packet test

From June to August 2020, adult female *R. annulatus* ticks were collected from naturally infected cattle visiting veterinary facilities and farms in the Fayoum governorate (hot seasons). The ticks were taken to the Faculty of Veterinary Medicine's Parasitology Laboratory at Beni- Suef University in Egypt. Tick samples were cleaned in distilled water and dried on filter paper before being identified using a stereobinocular microscope, weighted, and split into 10 groups. Adult female ticks were maintained in a BOD incubator until they produced the enormous quantity of eggs required for larval bioassays (14–18 days).

## Evaluation of NPs efficacy for removal of deltamethrin

**Larval bioassay (larval packet test) using freshly absorbed deltamethrin nanocomposites and after 48 h post adsorption.** The produced nanomaterials were administered to tick larvae via a larval package test after adsorption of deltamethrin in fresh form and after 48 hours post adsorption of deltamethrin to validate the elimination of deltamethrin from water. In a Petri plate, filter papers were placed, and one mL of each produced D and/or DNPs solution was placed on the filter sheets. After allowing the impregnated sheets to dry, they were folded into packages. A brush was used to put about 100 larvae into each experimental package, which was then sealed with bulldog clips. For 24 h, the treated packets were maintained in a controlled environment room at 26–28˚C and 80% relative humidity. Nanomaterial solutions were replaced with distilled water in the control group. After allowing the impregnated sheets to dry, they were folded to make packages. Each experimental package was sealed with bulldog clips after the larvae (about 100) were transferred using a brush. For 24 hours, the treated packets were maintained in a con- trolled environment chamber with a temperature of 26–28˚C and a humidity of 80%. Distilled water was used instead of nanomaterial solutions in the control group [39].

## Comet assay using deltamethrin nanocomposites and deltamethrin alone on treated tick larvae

Adult ticks that were still alive 24 hours after treatment were used in this test. The prescribed dosage (X) of D, free nanomaterials, and DNCs were applied to adult *R. annulatus* female ticks

using the adult immersion method. Ticks were submerged in 10 mL of the solution for 2 min (10 ticks per treatment), dried, and incubated in BOD on Petri dishes for 24 hours [40]. The modified single-cell gel electrophoresis or comet test was used on all tick tissues in the control and treatment groups [41]. Small portions of the tissues were rinsed in an ice-cold Hank's balanced salt solution (HBSS) and minced into fine bits, about one mm$^3$ pieces, using stainless steel scissors to extract the cells. The chopped tissues were then washed several times in cold phosphate-buffered saline before being pipette distributed into single cells. The comet test was performed by embedding whole tick cells in agarose and stacking them on microscope slides. The comet assay protocol's analysis and follow-up procedures were carried out as previously described [42]. The proportion of DNA damage in the tail of each tick's comet was determined by analyzing 100 cells from each tick. In each trial, non-overlapping cells were chosen at random and rated (0–3). Based on perceived comet tail length migration and relative proportion of DNA in the nucleus, score 0 = no detectable DNA damage and no tail; score 1 = tail with a length less than the diameter of the nucleus; score 2 = tail with a length between 1× and 2× the nuclear diameter; and score 3 = tail longer than 2 the diameter of the nucleus [42].

## Statistical analysis

To see whether factors varied across nanomaterials, tick biological characteristics were statistically evaluated using the Statistical Package for Social Science (SPSS for Windows (IBM), version 22, Chicago, USA). ANOVA tests were also used to evaluate the differences between the means. The results are shown as mean ± SE. $P$-values less than 0.05 ($P < 0.05$) were deemed significant.

## Results

### Characterization of the prepared nanoparticles

**FE-SEM and HR-TEM.** FE-SEM was used to investigate the morphology of the synthesized materials (Fig 1). Fig 1 depicts layer and sheet nanostructures. Fig 1 shows FE-SEM images of Zn-Al LDH/GA and Zn-Fe LDH that show a well-defined sheet structure. In the case of Zn-Fe LDH, it is loose sheets. In the case of the Zn-Al LDH/GA, however, the layers are compressed. The as-prepared Fe-oxide NPs were analyzed, and they clearly indicate the production of diverse spherical and compact tiny layer nanoparticles with various other shaped structures. HRTEM was used to characterize the As-synthesized NPs for microstructural investigation (Fig 1). Fig 1 depicts a typical HRTEM micrograph of LDH as layers and sheets or spherical-like Fe oxide nanoparticles (Fig 1) [43, 44].

**X-ray diffraction.** The XRD patterns of the produced Zn-Fe LDH were comparable to those of hydrotalcite-like LDH materials. The Zn-Fe LDH was extremely crystalline and had distinct diffraction peaks. The appearance of major peaks at 31.86˚, 34.6˚, 36.4˚, and 47.62˚ corresponding to the (003) plane proved the layered structure of Zn-Fe LDH. The comparison of XRD before and after deltamethrin adsorption showed a reduction in the strength of certain diffraction peaks, such as 34.55˚, 36.47˚, 47.41˚, and 68.04˚, as well as a shift of other peaks, such as 31.8˚ to 32.0˚, 56.7˚ to 57.06˚, and 62.9˚ to 63.04˚. The diffraction angle in the XRD pattern after conjugation was 23.7˚ and 59.7˚ and corresponds to deltamethrin. The deltamethrin/Zn-Fe LDH had a basal peak (2˚ = 36.4755˚) that corresponded to an interlayer-layer distance of 2.46337 A, which was 0.00047 A higher than the Zn-Fe LDH. This implies that deltamethrin was not intercalated into the Zn-Fe LDH layers, but the high-intensity peaks of 2 values (36.4827˚ and 36.4755˚) for Zn-Fe LDH and deltamethrin/Zn-Fe LDH, respectively, were caused by deltamethrin's interaction with metal cations, Zn(II) and Fe(III), of the LDH (Fig 2). The crystallinity of magnetite or hematite green clove synthetized Fe-oxide became

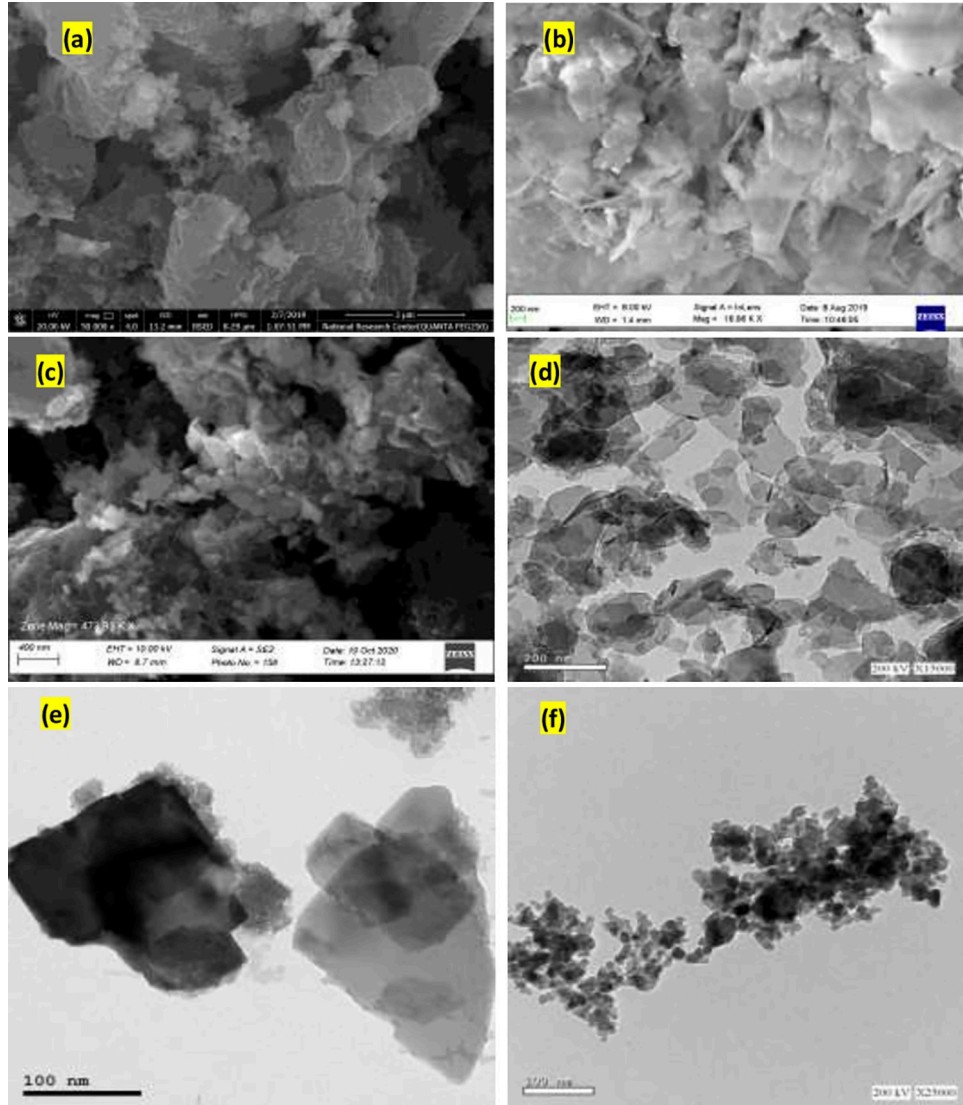

**Fig 1.** FESEM images of the prepared Zn-Al LDH/GA (a), Zn-Fe LDH (b) and Iron oxide (c) nanoparticles. HRTEM images for Zn-Al LDH/GA (d), Zn-Fe LDH (e) and Iron oxide (f).

apparent after calcination, and the majority of the produced sample changed to magnetite while the remainder turned to hematite. The calcinated green clove Fe-oxide diffraction peaks were matched to the hematite and magnetite XRD patterns in card numbers (04-015-9569 and 04-009-8420, respectively). The deltamethrin/Fe-oxide XRD pattern had mean diffraction peaks of 33.2˚, 35.69˚, 49.58˚, 54.13˚, and 64.08˚, which corresponded well with the hematite XRD pattern (04-015-9569), which had several low-intensity peaks, such as 35.6˚, 57.66˚, and 62.59˚. This corresponded to the magnetite XRD pattern (04-009-8420). The interaction of deltamethrin with Fe-oxide resulted in the conversion of the magnetite form of Fe-oxide to the hematite form of Fe-oxide. As a result, certain peaks had increased intensity (e.g., 33.2˚, 49.58˚, and 54.13˚), while others had reduced intensity (e.g., 35.69˚, 62.59˚). This change from magnetite to hematite happened because the magnetite form was less stable, causing agglomeration via magnetostatic interaction and oxygen adsorption (Fig 2).

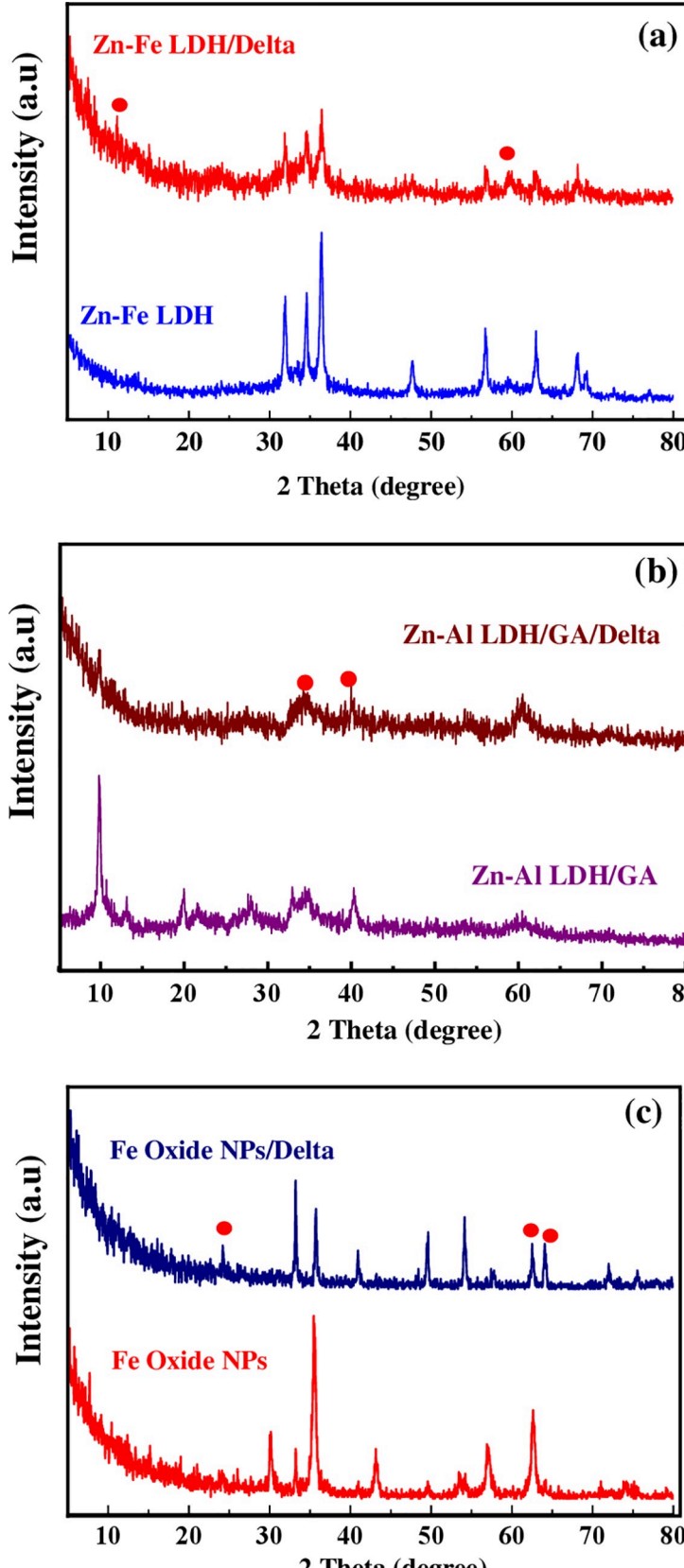

**Fig 2. XRD pattern of the as-synthesized nanomaterials compared to their pattern after adsorption of delta.**

**Fourier transform infrared (FT-IR).** The chemical interaction, as well as the alternation in chemical bonds, functional groups, and changes in the wave number of the peaks, verified the conjugation between deltamethrin and nanomaterials (Zn-Fe LDH, Zn-Al-GA LDH, and Fe-oxide). The bandwidth at 3477.72 cm$^{-1}$ was caused by the stretching vibrations of the hydroxyl groups in the layers and the interlayer water molecules. While the band at 1635.15 cm-1 is caused by water's bending vibration. Additionally, the band at 1377.95 cm-1 was matched to the nitrate ion's vibration mode. Bands less than 1000 cm$^{-1}$ were assigned to the vibration modes of Zn-O, Fe-O, O-Zn-O, Al-O, and O-Al-O (metal-oxygen bonds). The stretching vibration of C = C groups was assigned to 1499.77 cm-1, the stretching vibration of the hydroxyl group was assigned to 3396.04 cm-1, the bending vibration of the C-H was assigned to 834.89, 943.05, and 2920.81 cm-1, and the stretching vibration of a C-C bond was assigned to 1062.11 cm-1. The stretching vibration of the N-H bond produced a weak peak at 3731 cm-1. H-bonding formations were promoted by hydrogen donors (OH groups) on the surface of Zn-Fe LDH and hydrogen acceptors (-OH or -NH) in the deltamethrin structure. The deltamethrin interaction with Zn-Fe LDH was verified when the OH vibration mode peak changed from 3477 to 3396 cm$^{-1}$ (Fig 3). The Zn-Al-GA/LDH spectra revealed bands at 1546.98 cm-1 related to C = C, 1232.31 cm$^{-1}$ related to COO-, and 1036.59 cm-1 belonging to phenol groups. On Zn-Al-GA LDH, the characteristic peak of loaded deltamethrin was found at 2931.79 cm$^{-1}$, corresponding to aliphatic CH2 and CH, and at 692 cm$^{-1}$, corresponding to the in-plane bending vibration of replacement benzene, indicating the existence of C = C. The creation of hydrogen bonds between H bond donor oxygen atoms and LDH layers was verified by the rising intensity and shifting of peaks, indicating deltamethrin conjugation with Zn-Al-GA LDH (Fig 3). Fig 3 shows the FTIR spectra of Fe oxide before and after deltamethrin adsorption, with magnetite Fe-oxide absorbance bands at 686 and 598 cm$^{-1}$ and hematite Fe-oxide absorbance bands at 462 cm-1. The emergence of certain peak vibration modes at 1643 and 1734 cm$^{-1}$, as well as a reduction in the strength of vibration peaks at 457 and 540 cm$^{-1}$, suggested interaction of deltamethrin with Fe-oxide and conversion of magnetite to hematite Fe-oxide (Fig 3).

## Adsorption study

The pH has a great effect on the adsorption process (Fig 4). The isotherm models (Langmuir and Freundlich) described the adsorption process of deltamethrin (Table 2 and Fig 5 as a representative).

The kinetics of the adsorption test is crucial for the development of adsorbents because they introduce essential sources to the pollutants removal rate. Using kinetic models and predicting rate-controlling mechanisms such as diffusion control, mass transfer, and chemical reaction, the absorption of pollutants from a liquid phase by an adsorbent may be described. The experimental kinetic data, as well as the curves derived from the kinetic models, are shown in Fig 6. The equations and fitting parameters for the utilized kinetic models are given in (Tables 3–5).

## Evaluation of NPs efficacy for removal of deltamethrin

**Larval bioassay results of freshly adsorbed nanocomposites and after 48 h post-adsorption of deltamethrin.** At the indicated dosage, deltamethrin loaded Zn-Fe LDH, Fe-oxide, and Zn-Al-GA LDH nanoparticles induced larval mortality with no significant differences (P0.05) between deltamethrin alone and deltamethrin loaded Zn-Fe LDH, Fe-oxide, and Zn-Al-GA LDH nanoparticles. In addition, there were no significant changes in larval mortality between the X and D doses of DNPs and D alone, ranging from 79.6 to 85.6 percent (Table 6). The effectiveness of DNPs was evaluated using a packet test against tick larvae 48 h after

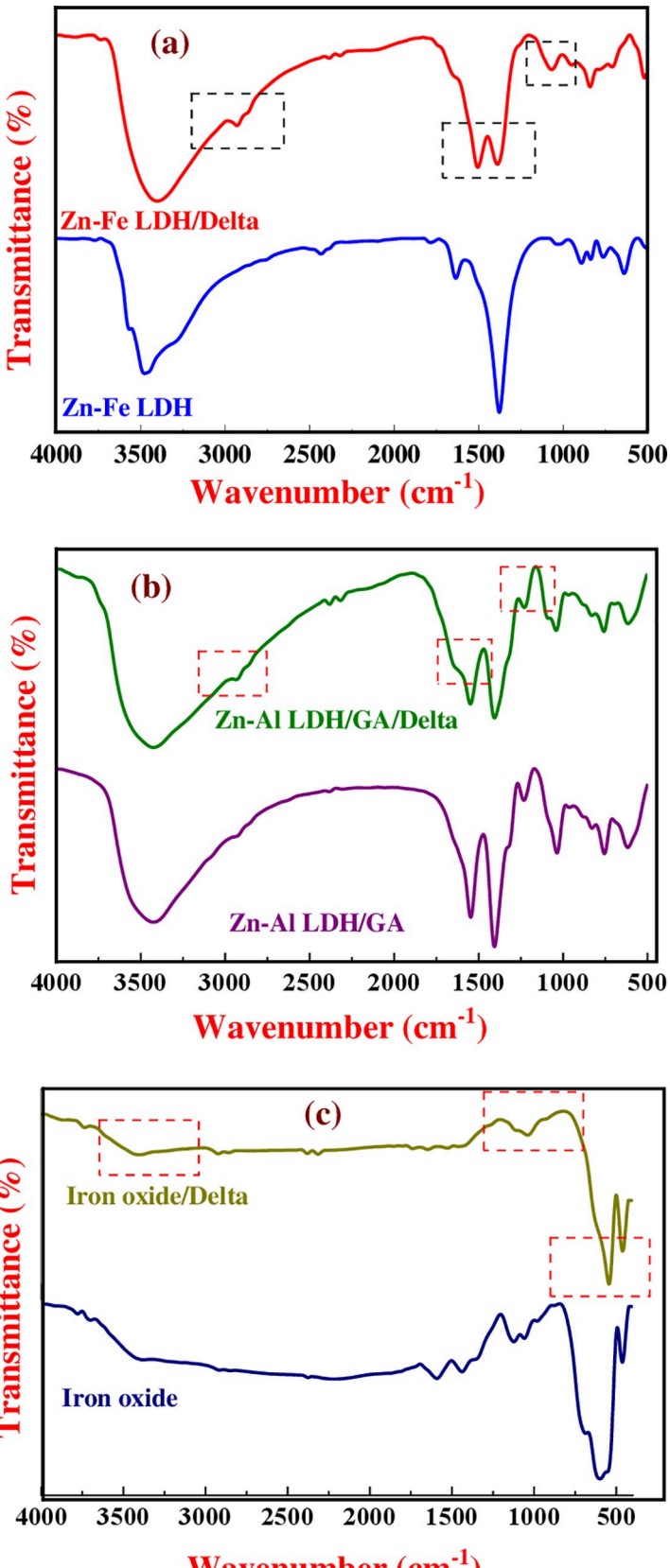

**Fig 3. FT1R spectra of the as-synthesized nanomaterials compared with their spectra after adsorption of deltamethrin.**

deltamethrin adsorption by nanomaterials. To ensure deltamethrin removal from water, the loaded DNPs must be produced and incubated for 48 hours before use. After 48 h of incubation with NPs, deltamethrin lost its acaricidal action, according to these tests. There is no significant difference in mortality between the tested larvae and the untreated control larvae. Deltamethrin alone, on the other hand, caused 75.66% larval death (Table 6).

## Comet assay

The comet test indicated 28.51±1.19% DNA damaged cells in deltamethrin-treated ticks, but 20.76±1.49, 17.79±0.85, and 22.76±1.38% DNA damaged cells in ticks treated with deltamethrin/Fe-oxide, deltamethrin/Zn-Fe LDH, and deltamethrin/Zn-Al-GA LDH, respectively (Table 7). The percentage of DNA damage in the soft tissues of treated ticks was scored from 0 to 3 in the control, deltamethrin alone, free nanomaterials, and deltamethrin-loaded nanomaterials, where score 0 represents normal cells and score 3 represents the most severely damaged cells, as shown in (Figs 7 and 8). Because of the development of an efficient Zn-Fe LDH conjugate with the greatest effective result in deltamethrin inactivation in water, the Zn-Fe LDH/deltamethrin had the least impact on DNA damage (Table 7). The comet test, UV-Vis analysis, and larval toxicity bioassay findings were all in good agreement. These findings show that conjugating deltamethrin with nanomaterials decreases deltamethrin's acaricidal action and allows deltamethrin to be inactivated by residual water.

## Discussion

The most widely used technique for pest control is synthetic chemical acaricides. The widespread use of deltamethrin, which was more likely to develop tick deltamethrin resistance, resulted in higher pollution and posed a greater threat to public health. Nanotechnology has the potential to address a variety of technical issues in a variety of fields, including agriculture, antimicrobial agents, medical transporters, and pesticides [45, 46]. The deltamethrin was absorbed by Zn-Fe LDH, Zn-Al-GA LDH, and Fe-oxide nanocomposites. The use of XRD, FT-IR, SEM, and TEM to demonstrate deltamethrin absorption and interaction with nanomaterials was accomplished. These methods for ensuring effective adsorption and characterization are similar to those previously described [34, 38, 44]. The interaction of deltamethrin with metal cations; Zn (II) and Fe (III) of the LDH, respectively, resulted in high-intensity peaks of 2θ values for Zn-Fe LDH-NPs and deltamethrin/Zn-Fe LDH-NPs. Furthermore, certain peaks in the deltamethrin/Zn-Al-GA/LDH-NPs XRD pattern grew wide, which may be owing to the presence of deltamethrin, which could induce exfoliation of LDH layers [47, 48]. The interaction of deltamethrin with Fe-oxide resulted in the conversion of the magnetite form of Fe-oxide to the hematite form. As a result, the strength of certain peaks rose, while others dropped. The magnetite state was less stable, thus the change from magnetite to hematite occurred. Magnetostatic interaction and oxygen adsorption cause agglomeration in this way [49, 50]. Furthermore, the FT-IR spectra of produced Zn-Fe LDH matched those of previous studies [34, 51]. The deltamethrin interaction with Zn-Fe LDH was verified when the OH vibration mode peak changed from 3477 to 3396 cm$^{-1}$ [44, 52]. The creation of hydrogen bonds between H bond donor oxygen atoms and LDH layers is verified by the conjugation of deltamethrin with Zn-Al-GA LDH, as shown by the rise in intensity and shifting of peaks in the FT-IR spectra of Zn-Al-GA LDH/deltamethrin [35, 53]. The appearance of some peak

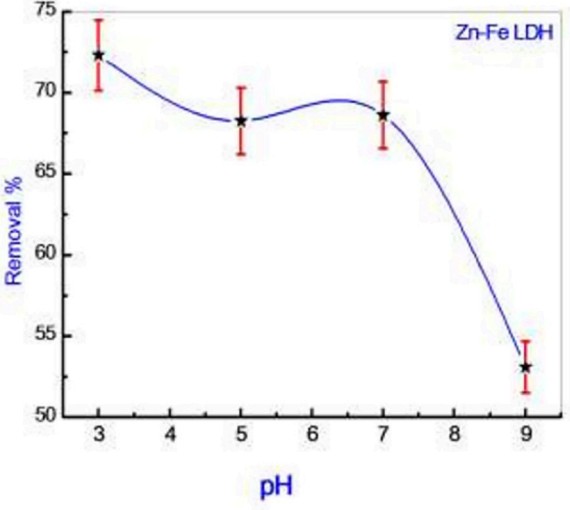

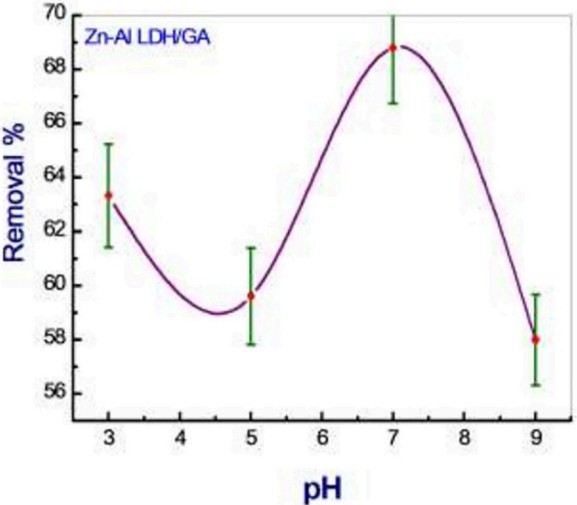

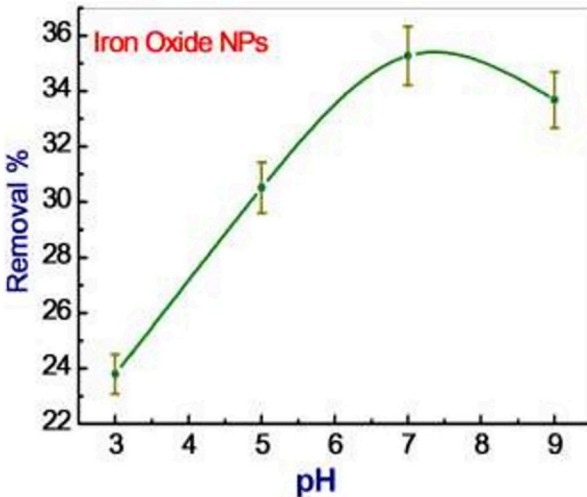

**Fig 4. Percent removal of different nanomaterials at different pH values.**

**Table 2. Adsorption isotherm constants for the adsorption of deltamethrin in aqueous system systems.**

| Adsorbent Isotherm models | Adjustable model parameters | Values | $R^2$ |
|---|---|---|---|
| Zn-Fe LDH | | | |
| Langmuir | $q_{max}$ | 56.80 | 0.97 |
| | $K_{ad}$ | 0.043 | |
| Freundlich | $K_f$ | 5.15 | 0.97 |
| | $1/n_F$ | 0.53 | |
| Zn-Al LDH/GA | | | |
| Langmuir | $q_{max}$ | 47.44 | 0.99 |
| | $K_{ad}$ | 0.02 | |
| Freundlich | $K_f$ | 1.80 | 0.99 |
| | $1/n_F$ | 0.68 | |
| Iron Oxide NPs | | | |
| Langmuir | $q_{max}$ | 20.36 | 0.98 |
| | $K_{ad}$ | 0.043 | |
| Freundlich | $K_f$ | 2.22 | 0.917 |
| | $1/n_F$ | 0.476 | |

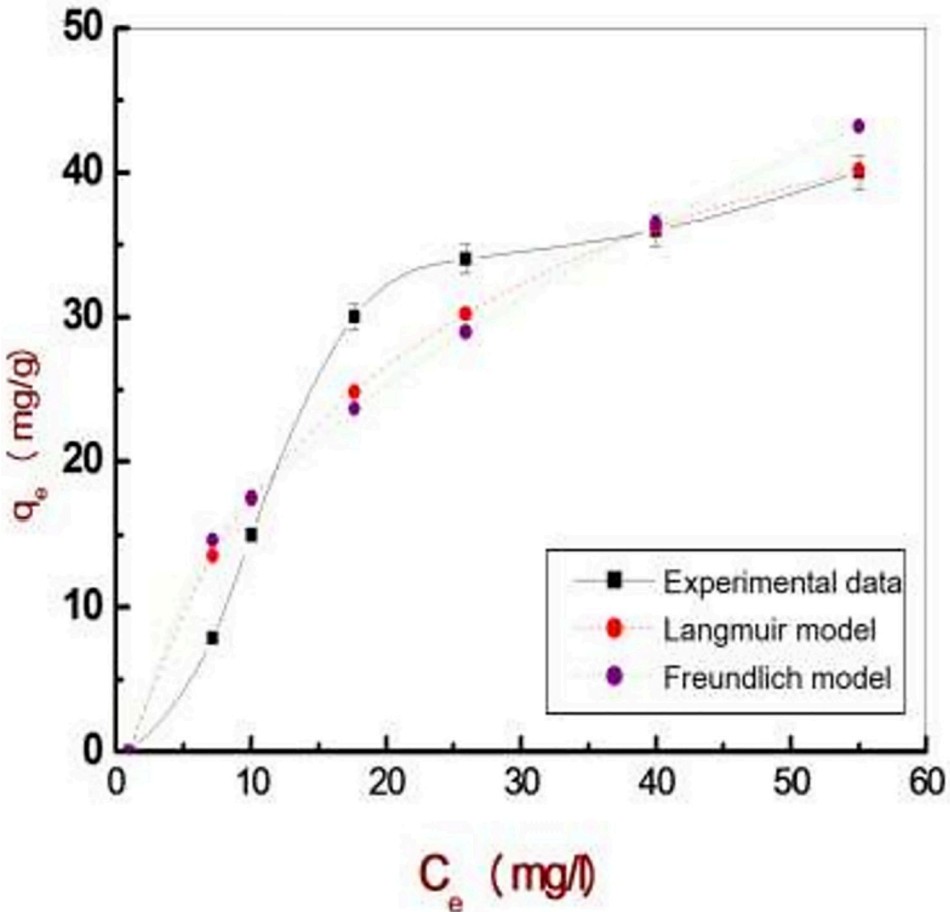

**Fig 5. Experimental adsorption isotherm data of deltamethrin on the Zn-Fe LDH fitted using the two-parameter isotherm.**

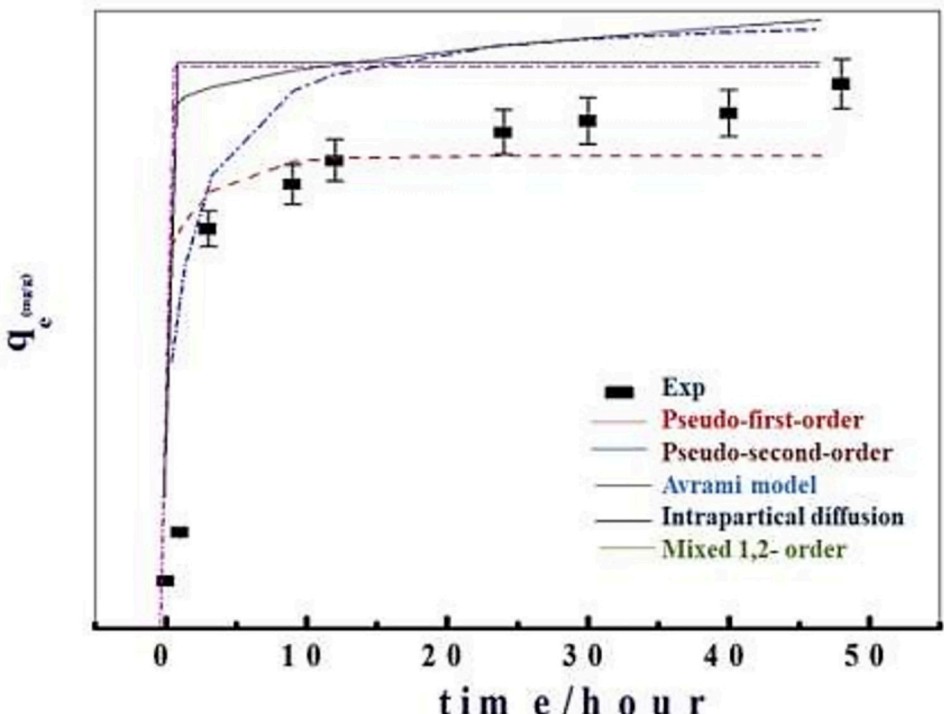

**Fig 6. Fitting of the experimental data at Co 40 mg/L of pseudo 1st order, pseudo 2nd order, intraparticle diffusion, mixed 1st & 2nd orders and Avrami for the experimental data of the deltamethrin adsorption onto nanoparticles.**

vibration modes at 1643 and 1734 cm$^{-1}$ decreases the intensity of vibration peaks at 457 and 540 cm$^{-1}$ in the FT-IR spectrum of deltamethrin/Fe-oxide, indicating the interaction of delta-methrin with Fe-oxide and the conversation of magnetite to hematite Fe-oxide, as well as the disappearance of a peak at 686 cm$^{-1}$ [54, 55].

**Table 3. The adsorption kinetic models for deltamethrin adsorption using Zn-Al LDH/GA nanoparticles and their parameters obtained from the fitting results.**

| Kinetic models | $R^2$ | Model parameters | Zn-Al DH/GA |
|---|---|---|---|
| pseudo-first-order $q_t = q_e(1 - e^{-k_1 t})$ | 0.994 | $k_1$ (min$^{-1}$) | 1000 |
| | | $q_{(e,cal)}$ (mg/g) | 31.25 |
| | | $q_{(e, exp)}$ (mg/g) | 32.12 |
| pseudo-second-order $q_t = \frac{q_e^2 k_2 t}{1 + q_e k_t t}$ | 0.996 | $k_2$ (g/mg/min) | $10^6$ |
| | | $q_{(e,cal)}$ (mg/g) | 30.918 |
| | | $q_{(e, exp)}$ (mg/g) | 32.12 |
| Intraparticle diffusion $q_t = k_{ip}\sqrt{t} + c_{ip}$ | 0.01 | $k_{ip}$(mg/g·min$^{0.5}$) | 0.46 |
| | | $c_{ip}$ (mg/g) | 29.05 |
| Avrami model $q_t = q_e(1 - e^{(-k_{av}t)^{n_{av}}})$ | 0.999 | $q_{(e,cal)}$ mg/g | 31.50 |
| | | $k_{av}$ (min$^{-1}$) | 0.84 |
| | | $n_{av}$ | 0.80 |
| Mixed 1, 2-order $q_t = q_e \frac{1 - \exp(-kt)}{1 - f_2 \exp(-kt)}$ | 0.999 | $k$ (mg·g$^{-1}$·min$^{-1}$) | 0.0011 |
| | | $q_{(e,cal)}$ mg/g | 32.36 |
| | | $q_{(e,exp)}$ mg/g | 32.12 |
| | | $f_2$ | 0.999 |

**Table 4. The adsorption kinetic models for deltamethrin adsorption using Zn-Fe LDH nanoparticles and their parameters obtained from the fitting results.**

| Kinetic models | $R^2$ | Model parameters | Zn-Fe/LDH |
|---|---|---|---|
| pseudo-first-order $q_t = q_e(1 - e^{-k_1 t})$ | 0.980 | $k_1$ (min$^{-1}$) | 10000 |
| | | $q_{(e,cal)}$ (mg/g) | 26.065 |
| | | $q_{(e, exp)}$ (mg/g) | 29.00 |
| pseudo-second-order $q_t = \frac{q_e^2 k_2 t}{1 + q_e k_2 t}$ | 0.965 | $k_2$ (g/mg/min) | $1.03*10^6$ |
| | | $q_{(e,cal)}$ (mg/g) | 29 |
| | | $q_{(e, exp)}$ (mg/g) | 29.00 |
| Avrami model $q_t = q_e(1 - e^{(-k_{av}t)^{n_{av}}})$ | 0.999 | $q_{(e,cal)}$ mg/g | 28.64 |
| | | $k_{av}$ (min$^{-1}$) | 1.004 |
| | | $n_{av}$ | 0.949 |
| | | $q_{(e, exp)}$ (mg/g) | 0.999 |
| Mixed 1, 2-order $q_t = q_e \frac{1 - \exp(-kt)}{1 - f_2 \exp(-kt)}$ | 0.998 | $k$ (mg·g$^{-1}$·min$^{-1}$) | 0.0019 |
| | | $q_{(e,cal)}$ mg/g | 29.71 |
| | | $f_2$ | 0.999 |

The pH has a great effect on the adsorption process. The Deltamethrin removal sharply increased to 68.91%, 68.54% and 35.32% using Zn-Al LDH/GA, Zn-Fe LDH and iron oxide NPs, respectively, at a pH of 7. At a pH higher beyond pH 9, the adsorption of deltamethrin likely decreased owing to OH− groups that are repulsed to the negative molecules of deltamethrin. Isotherm models explain the behavior of the adsorption of deltamethrin on three nanomaterials well upon comparing the calculated values from adsorption isotherms with experimental values applied to fit the experimental data using a nonlinear relationship with a Langmuir adsorption isotherm model [56]. The Langmuir adsorption isotherm is widely used for the modeling of homogeneous adsorption on the surface of the monolayer and assumes that the adsorbent surface is uniform and that all sorption sites are identical. The Freundlich isotherm model is suitable for heterogeneous isotherm model is used for both heterogeneous and homogeneous distributions at high and low concentrations. The results show the

**Table 5. The adsorption kinetic models for deltamethrin adsorption using Fe-oxide nanoparticles and their parameters obtained from the fitting results.**

| Kinetic models | $R^2$ | Model parameters | Fe- oxide |
|---|---|---|---|
| pseudo-first-order $q_t = q_e(1 - e^{-k_1 t})$ | 0.833 | $k_1$ (min$^{-1}$) | 1000 |
| | | $q_{(e,cal)}$ (mg/g) | 24.36 |
| | | $q_{(e, exp)}$ (mg/g) | 31.35 |
| pseudo-second-order $q_t = \frac{q_e^2 k_2 t}{1 + q_e k_2 t}$ | 0.806 | $k_2$ (g/mg/min) | $1.03*10^6$ |
| | | $q_{(e,cal)}$ (mg/g) | 24.36 |
| | | $q_{(e, exp)}$ (mg/g) | 31.35 |
| Intraparticle diffusion $q_t = k_{ip}\sqrt{t} + c_{ip}$ | 0.436 | $k_{ip}$ (mg/g·min$^{0.5}$) | 1.886 |
| | | $c_{ip}$ (mg/g) | 18.103 |
| Avrami model $q_t = q_e(1 - e^{(-k_{av}t)^{n_{av}}})$ | 0.983 | $q_{(e,cal)}$ mg/g | 28.92 |
| | | $k_{av}$ (min$^{-1}$) | 0.579 |
| | | $n_{av}$ | 0.547 |
| Mixed 1, 2-order $q_t = q_e \frac{1 - \exp(-kt)}{1 - f_2 \exp(-kt)}$ | 0.995 | $k$ (mg·g$^{-1}$·min$^{-1}$) | 0.0019 |
| | | $q_{(e,cal)}$ mg/g | 32.15 |
| | | $f_2$ | 0.999 |

**Table 6. Larval mortality percentage as an indicator of deltamethrin removal by the freshly adsorbed deltamethrin/nanomaterials and after 48 h post-adsorption with nanomaterials.**

| Deltamethrin, deltamethrin/nanomaterials (Deltamethrin (X) = 1 uL/mL) | Larval mortality percentage (%) by application of freshly adsorbed deltamethrin/nanomaterials | Larval mortality percentage (%) by application of adsorbed deltamethrin/nanomaterials 48hrs post adsorption |
|---|---|---|
| Deltamethrin (1 uL/mL) | 79.7 ± 0.8 | 79.7 ± 0.8 |
| Zn-Fe/LDH | 9.83± 0.5 | 9.83± 0.5 |
| Zn-AL-GA/LDH | 11.2 ± 0.8 | 11.2 ± 0.8 |
| Fe-oxide | 10.2 ± 1.1 | 10.2 ± 1.1 |
| Deltamethrin/Zn-Fe LDH | 79.0 ± 0.6 | 10.0 ± 1.1* |
| Deltamethrin/Zn-Al-GA LDH | 84.0 ± 1.0 | 9.50 ± 1.0* |
| Deltamethrin/Fe-oxide | 85.7 ± 0.4 | 10.0 ± 1.4* |
| Control (distilled water) | 7.17± 0.5 | 7.17± 0.5 |

(*) Significant $P \leq 0.05$.

adsorption behavior of deltamethrin well based on the statistical analysis of the correlation coefficient R2; whereas the qe was 56.80, 47.44 and 20.36 mg/g for Zn-Fe LDH, Zn-Al LDH/ GA and Iron oxide NPs. Based upon this result, the Langmuir model was the best model for explaining the adsorption process, where homogeneous adsorption is on the surface of the monolayer, and the surface of adsorbent is uniform and without interactions between adsorbents. This indicates that the Langmuir model is more suitable for explaining the process of deltamethrin adsorption and better represents the experimental data.

The experimental kinetic data revealed fast absorption of deltamethrin during the first 1 hour, followed by gradual elimination until equilibrium was reached at 24 hours, and then stability was achieved for up to 48 hours. The adsorption kinetic data were best matched with the pseudo-first-order, pseudo-second-order, Avrami model, and the mixed 1, 2-order for the three adsorption systems, according to the error function correlation coefficient R2 (Tables 3–5). The predicted qe (q(e,cal)) is extremely similar to the experimentally determined equilibrium q(e,exp) (Tables 3–5). As a result, the avrami and mixed 1, 2 order adsorption mechanisms are dominant for the adsorption of deltamethrin onto Zn-Al LDH/GA LDH, Zn-Fe LDH, and Fe-oxide nanoparticles, implying that physical adsorption and intermolecular intermolecular hydrogen bonding and chemical bonds were the rate determining steps through electrostatic interactions and intermolecular [57].

**Table 7. Comet assay using deltamethrin, nanocomposites and its loading forms on treated tick larvae.**

| Treatment | Number of ticks | No. of cells | | Class¥ of comet | | | | DNA damaged cells (%) (Mean ± SEM) |
|---|---|---|---|---|---|---|---|---|
| | | Analyzed (*) | Total comets | 0 (Normal cells) | 1 | 2 | 3 | |
| Control (70% Ethanol) | 4 | 400 | 31 | 369 | 28 | 3 | 0 | 7.75±0.63 |
| Deltamethrin | 4 | 400 | 114 | 286 | 35 | 42 | 37 | 28.51±1.19 |
| Fe–oxide | 4 | 400 | 47 | 353 | 33 | 14 | 0 | 11.78±1.38 |
| Fe-oxide/ deltamethrin | 4 | 400 | 83 | 317 | 30 | 32 | 21 | 20.76±1.49 |
| Zn-Fe LDH | 4 | 400 | 42 | 358 | 25 | 13 | 4 | 10.52±1.04 |
| Zn-Fe LDH/Deltamethrin | 4 | 400 | 71 | 329 | 31 | 27 | 13 | 17.79±0.85 |
| Zn-Al-GA LDH | 4 | 400 | 56 | 344 | 33 | 18 | 5 | 14.12±0.91 |
| Zn-Al-GA LDH/Deltamethrin | 4 | 400 | 91 | 309 | 37 | 25 | 29 | 22.76±1.38 |

¥: Class 0 = no tail; 1 = tail length < diameter of nucleus; 2 = tail length between 1X and 2X the diameter of nucleus; and 3 = tail length > 2X the diameter of nucleus.(*): No of cells analyzed were 100 per a tick.

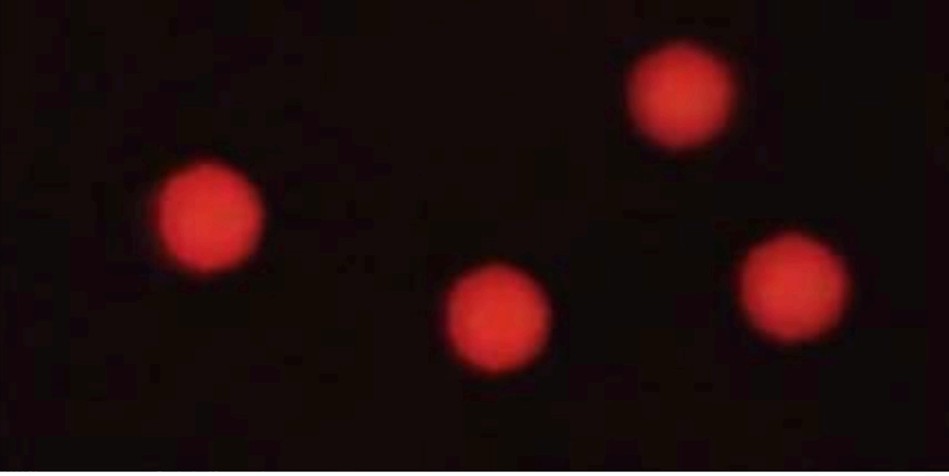

**Visual score of normal DNA (class 0) using comet assay in soft tissue samples collected from the control ticks**

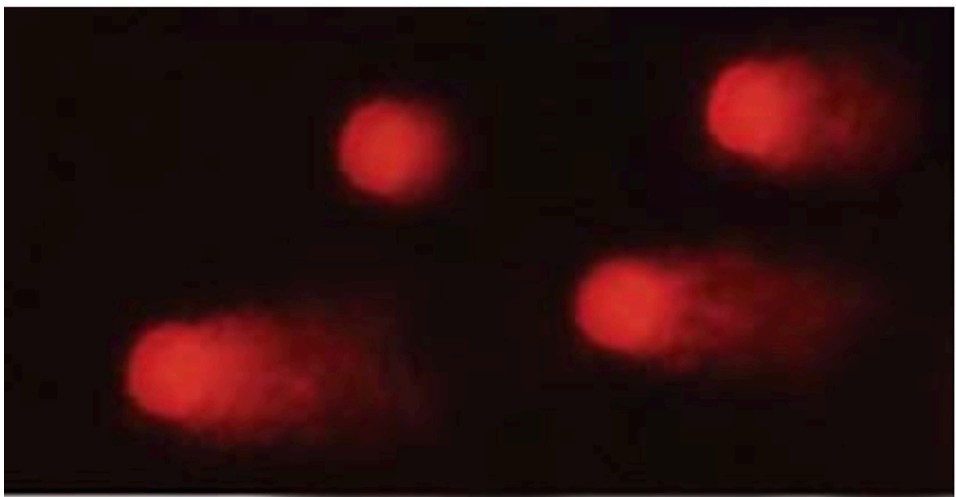

**Fig 7. Visual score of damaged DNA (class 1, class 2 & class 3) using comet assay in soft tissue samples collected from the treated ticks.**

NPs were used to investigate the effective removal of deltamethrin from water. At the concentration (X) of deltamethrin and its newly adsorbed DNPs forms, the larval toxicity bioassay revealed 79.7 ± 0.8% death. The impact of deltamethrin and newly adsorbed DNPs on annulatus larvae was not significantly different ($p{\geq}0.05$). This shows that the nanocomposites have no impact on the effectiveness of deltamethrin in its newly adsorbed state. Using DNPs after 48 h of adsorption; however, resulted in non-significant mortality of larvae as compared to control untreated larvae. At the concentration (X) of deltamethrin, however, deltamethrin alone caused 79.7% of larval death. As a result, this discovery proved deltamethrin's removal from water and the lack of its acaricidal action. As adsorbent nanoparticles for deltamethrin contamination on wastewater cleanup, Zn-Fe/LDH, Zn-Al-GA/LDH, and Fe-oxide were utilized. After 48 h of incubation at room temperature, these NPs successfully removed deltamethrin from the water. The drug residues were removed from water using these adsorbent nanoparticles [58]. UV-Vis spectrophotometer, GCMS analysis, and FT-IR analysis were used

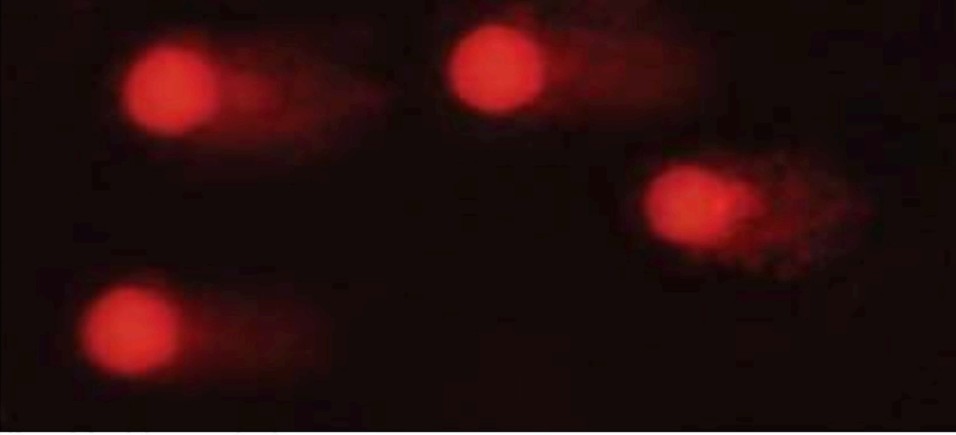

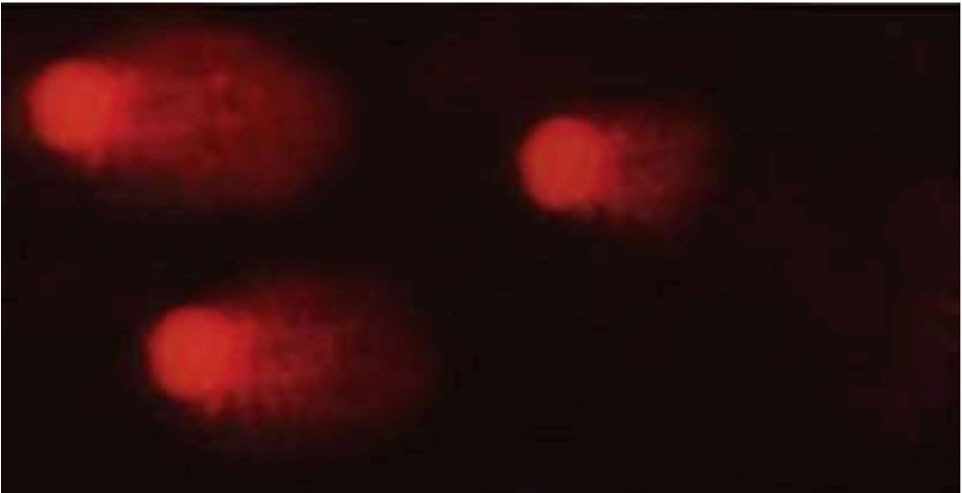

**Fig 8. Visual score of damaged DNA (class 2 and class 3) using comet assay in soft tissue samples collected from the treated ticks.**

to demonstrate deltamethrin breakdown by the bacterial isolate (IK2a) [3]. The degradation of deltamethrin differed very slightly amongst the nanoparticles utilized. These results align with those of [37], who utilized magnetic iron oxide nanoparticles to extract deltamethrin from water. Furthermore, in the physicochemical treatment by ion exchange and the precipitation technique, the pesticides were adsorbed in two ways [59, 60]. The present finding is corroborated by the findings of [44], who discovered that Zn-Fe LDH is capable of effectively removing oxytetracycline hydrochloride. According to them, this technique is low-cost and creates bonds between oxytetracycline molecules and Zn, as well as Fe atoms of LDH forming hydrogen bonds with the hydroxyl groups of LDH. Furthermore, iron oxide nanoparticles exhibit unique paramagnetic properties that promote cellular and molecular interactions. Because of these characteristics, Fe-oxide may be employed as a contrast agent in disease diagnostics and as a medication carrier [61, 62]. Zn-Fe LDH was used to remove a large amount of cadmium from waste for environmental cleanup. At the same time, Zn-Fe LDH is an antibacterial agent

that works against both Gram-positive and Gram-negative bacteria. As a result, Zn-Fe LDH has a lot of potential in the area of water bioremediation [34]. The interchange of phosphorus and the hydroxyl group on the surface of magnetic LDHs (Mg-Al, Zn-Al, and Mg-Fe LDHs) makes them good adsorbents for phosphorus [63]. In terms of environmental remediation, Zn-Fe LDH removed a considerable amount of cadmium from waste. Zn-Fe LDH is also a potent antibacterial agent against Gram-positive and Gram-negative bacteria. As a result, Zn-Fe LDH has a lot of potential in the area of water bio-remediation [34]. The interchange of phosphorus and the hydroxyl group on the surface of magnetic LDHs (Mg-Al, Zn-Al, and Mg-Fe LDHs) makes them good phosphorus adsorbents [64]. Recent research has looked at the dual function of LDH nanoparticles. LDH nanoparticles are adsorbent, nontoxic, fast, and easy to produce, with properties comparable to clay materials, and are effective in removing drug residues from water. As an adsorbent, Zn-Fe LDH has been shown to effectively remove a high proportion of diclofenac sodium from an aqueous solution [38]. Sulfamethoxazole drug removal capability by calcined LDH under optimum circumstances with a 93% elimination rate was reported by [64]. These findings may be helpful in the battle against antibiotic resistance in bacteria for therapeutic reasons. Copper ions and malachite green dye were also eliminated by Zn-Al-GA LDH/polystyrene nanofibers [53]. Furthermore, Zn-Fe LDH showed the capacity to adsorb arsenic and antimony from contaminated water [65]. After a 24-hour incubation period on the soft tissues of treated ticks, the comet test was used to determine the in vitro genotoxic activity of commercial deltamethrin, free nanomaterials, and deltamethrin nanomaterials complex. The comet test revealed that the treated groups had more DNA damage than the control group. Furthermore, the treated groups exhibited significant drug concentrations in their tissues, suggesting a link between DNA damage and drug exposure [66–68]. The comet assay revealed that the rate of DNA damage and apoptosis in soft tissues of ticks treated with Zn-Fe LDH/deltamethrin, Zn-Al-GA LDH/deltamethrin, and Fe-oxide/deltamethrin was lower than the rate of DNA damage in cells treated with deltamethrin alone, which was in agreement with our findings. When compared to free nanomaterials, the untreated control group (score 0), and deltamethrin loaded nanomaterials, which produced a low rate of DNA damage, deltamethrin substantially induced DNA damage with an increased comet tail length (score 3). Deltamethrin's genotoxic activity and apoptotic impact on normal soft tissues may be reduced by using nanomaterials [69]. Furthermore, in DNA damaged cells, Zn-Fe LDH was the lowest, suggesting that this material absorbed the most deltamethrin from the water. The applied nanoparticles' mode of action in the deactivation of deltamethrin in dilute water may be ascribed to the nanomaterials' ability to effectively remove deltamethrin from water. Deltamethrin conjugation with nanomaterials also increased with time. The larval toxicity test showed that deltamethrin-loaded nanocomposites had no acaricidal activity. Furthermore, the comet test revealed that deltamethrin-loaded nanocomposites exhibited lower larval toxicity on DNA-damaged cells than deltamethrin alone. Finally, the findings of the Kinetic study of adsorption test, larvae toxicity bioassay, and comet assay were all compatible.

## Conclusion

The capacity of nanoparticles, Zn-Fe LDH-, Zn-Al-GA LDH-, and Fe-oxide to decontaminate deltamethrin was examined in the present research as a recent development in potential environmental remediation. After 48 hours of incubation at room temperature, these nanoparticles (Zn-Fe LDH, Zn-Al-GA LDH, and Fe-oxide) can remove deltamethrin from the water. Kinetic adsorption studies, larval toxicity bioassays, and comet assays all confirmed this elimination. As a result, these nanoparticles provide an alternate approach for substantial deltamethrin decontamination and environmental cleanup.

## Acknowledgments

The authors appreciate the help of veterinary clinics in conducting of this study.

## Author Contributions

**Conceptualization:** Samar M. Ibrahium, Ahmed A. Wahba, Shawky M. Aboelhadid.

**Data curation:** Samar M. Ibrahium, Ahmed A. Farghali, Rehab Mahmoud, Ahmed A. Wahba, Saeed El-Ashram.

**Formal analysis:** Samar M. Ibrahium, Shawky M. Aboelhadid.

**Funding acquisition:** Samar M. Ibrahium, Shawky M. Aboelhadid.

**Investigation:** Samar M. Ibrahium, Hesham A. Mahran, Shawky M. Aboelhadid.

**Methodology:** Samar M. Ibrahium, Ahmed A. Farghali, Rehab Mahmoud, Saeed El-Ashram, Shawky M. Aboelhadid.

**Project administration:** Samar M. Ibrahium, Ahmed A. Farghali, Shawky M. Aboelhadid.

**Resources:** Samar M. Ibrahium, Ahmed A. Farghali, Rehab Mahmoud, Saeed El-Ashram, Shawky M. Aboelhadid.

**Software:** Samar M. Ibrahium, Shawky M. Aboelhadid.

**Supervision:** Ahmed A. Farghali, Ahmed A. Wahba, Shawky M. Aboelhadid.

**Validation:** Shawky M. Aboelhadid.

**Visualization:** Samar M. Ibrahium, Shawky M. Aboelhadid.

**Writing – original draft:** Samar M. Ibrahium, Rehab Mahmoud, Shawky M. Aboelhadid.

**Writing – review & editing:** Samar M. Ibrahium, Saeed El-Ashram, Hesham A. Mahran, Shawky M. Aboelhadid.

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
