## [Decision Letter · Decision Letter 0]

31 Aug 2021

PONE-D-21-25490

New insight on some selected nanoparticles as an effective adsorbent toward diminishing the health risk of deltamethrin contaminated water

PLOS ONE

Dear Dr. ibrahium,

Thank you for submitting your manuscript to PLOS ONE. After careful consideration, we feel that it has merit but does not fully meet PLOS ONE’s publication criteria as it currently stands. Therefore, we invite you to submit a revised version of the manuscript that addresses the points raised during the review process.

We look forward to receiving your revised manuscript.

Kind regards,

Amitava Mukherjee, ME, Ph.D.

Academic Editor

PLOS ONE

Journal Requirements:

Reviewers' comments:

Reviewer's Responses to Questions

**Comments to the Author**

1. Is the manuscript technically sound, and do the data support the conclusions?

Reviewer #1: Yes

2. Has the statistical analysis been performed appropriately and rigorously? 

Reviewer #1: Yes

3. Have the authors made all data underlying the findings in their manuscript fully available?

Reviewer #1: Yes

4. Is the manuscript presented in an intelligible fashion and written in standard English?

Reviewer #1: Yes

5. Review Comments to the Author

Reviewer #1: The manuscript deals with the "removal of insecticide with adsorption methods (nanoparticles adsorbents).

The movement of pesticides into water bodies can occur through sub-surface drainage, leaching, run-off and spray drift. Hence, the elimination of pesticide residues from the environment has gained widespread attention. Insecticides are broadly applied to control insect pests, but their environmental safety has been a serious concern. The global insecticide market accounted for $14.51 billion in 2016, and it is forecasted to reach $19.27 billion by 2022.

1. Page 2; "..several kinetic models at pH 7, 40 ppm and 25 °C." It should be corrected as "several kinetic models at pH 7, initial concentration of deltamethrin 40 ppm and temperature 25 °C.

2. Page 2, "..with removal of 27.56 mg/g, 17.60 mg/g, and 3.06 mg/g of Zn-Al LDH/GA, Zn-Fe LDH, and Fe Oxide, respectively."?! What's the meaning here? It means:

deltamethrin removal of 27.56 mg/L, 17.60 mg/L, and 3.06 mg/L with the Zn-Al LDH/GA, Zn-Fe LDH, and Fe Oxide, respectively.

OR

in deltamethrin removal, the maximum adsorption capacity was 27.56 mg/L, 17.60 mg/L, and 3.06 mg/L with the Zn-Al LDH/GA, Zn-Fe LDH, and Fe Oxide, respectively.

3. Effects of pH on adsorption of insecticide should be testes, as an important factor in removal of organic pollutants. I suggest to use a wide range of pH (i.e. 3 to 9).

4. Testing adsorption isotherms, Langmuir and Freundlich, are recommended.

5. The main removal mechanism? electrostatic interactions? van der Waals?

6. PLOS authors have the option to publish the peer review history of their article (what does this mean?). If published, this will include your full peer review and any attached files.

Reviewer #1: No

---

## [Author Response · Author response to Decision Letter 0]

15 Sep 2021

Dear academic editor and reviewers, thank you very much for your valuable comments that help us in improving the manuscript.

---

## [Decision Letter · Decision Letter 1]

5 Oct 2021

New insight on some selected nanoparticles as an effective adsorbent toward diminishing the health risk of deltamethrin contaminated water

PONE-D-21-25490R1

Dear Dr. ibrahium,

We’re pleased to inform you that your manuscript has been judged scientifically suitable for publication and will be formally accepted for publication once it meets all outstanding technical requirements.

Kind regards,

Amitava Mukherjee, ME, Ph.D.

Academic Editor

PLOS ONE

Additional Editor Comments (optional):

Reviewers' comments:

Reviewer's Responses to Questions

**Comments to the Author**

1. If the authors have adequately addressed your comments raised in a previous round of review and you feel that this manuscript is now acceptable for publication, you may indicate that here to bypass the “Comments to the Author” section, enter your conflict of interest statement in the “Confidential to Editor” section, and submit your "Accept" recommendation.

Reviewer #1: All comments have been addressed

2. Is the manuscript technically sound, and do the data support the conclusions?

Reviewer #1: Yes

3. Has the statistical analysis been performed appropriately and rigorously? 

Reviewer #1: Yes

4. Have the authors made all data underlying the findings in their manuscript fully available?

Reviewer #1: Yes

5. Is the manuscript presented in an intelligible fashion and written in standard English?

Reviewer #1: Yes

6. Review Comments to the Author

Reviewer #1: The manuscript deals with the "removal of deltamethrin by a modified adsorption method".

The reviewers' comments have been addressed.

7. PLOS authors have the option to publish the peer review history of their article (what does this mean?). If published, this will include your full peer review and any attached files.

Reviewer #1: No

---

## [Editor Report · Acceptance letter]

26 Oct 2021

PONE-D-21-25490R1 

New insight on some selected nanoparticles as an effective adsorbent toward diminishing the health risk of deltamethrin contaminated water 

Dear Dr. Ibrahium:

I'm pleased to inform you that your manuscript has been deemed suitable for publication in PLOS ONE. Congratulations! Your manuscript is now with our production department. 

Kind regards, 

on behalf of

Professor Dr. Amitava Mukherjee 

Academic Editor

PLOS ONE